# METHODS FOR PATENT LANDSCAPE MODELLING AND PREDICTIVE ANALYSIS

## ABSTRACT

The article is dedicated to the problem of forecasting changes in the patent landscape based on the analysis of multimodal data. This article examines three main approaches to the analysis of patent information: 1) a clustering-based approach; 2) a resource-based approach; 3) a machine learning approach. Each approach is considered in terms of its advantages, flaws and potential for use in predicting the emergence of new technologies. The article proposes new methods for constructing a patent landscape and predictive models, as well as a method for assessing development directions in technological areas. The article also considers the problem of visualizing the results of analysis and forecasting of technological trends in order to provide a clear visual representation of the patent landscape. The article presents the results of experiments demonstrating the ability of the proposed methods to make correct predictions of technological trends.

**Keywords**—Clustering, machine learning, patent analysis, trend forecasting.

## 1 INTRODUCTION

Modern big data processing systems analyze multimodal data considering it as a combination of text, images, audio and video taken from different sources. This kind of analysis requires methods that effectively process data of heterogeneous formats and extract mutually complementary information from it.

Neural networks have proven their high effectiveness in complex and heterogeneous data processing tasks, solving them due to the ability to learn from large volumes of data and to identify the hidden patterns. Neural networks play a special role in the analysis of images (Anwar, 2018), texts (Tarwani & Edem, 2017) and graphs (Wu et al., 2020). There are many examples in the literature, and they include the following. The image analysis identifies diseases of human organs based on tomographic data (Anwar, 2018). The text data analysis identifies key topics and sentiments, as well as extracting data from unstructured texts (Tarwani & Edem, 2017). The graph data analysis identifies intricate relationships and dependencies, such as influential nodes, it detects communities, or finds the shortest paths to address real-world challenges in such areas as social network analysis, recommendation systems, fraud detection and so on (Wu et al., 2020).

The use of deep neural network architectures such as transformers increases the accuracy and depth of analysis considering the context and interrelations of the different elements (Gillioz, 2020). The use of neural networks for multimodal data analysis is of great importance in the development of intelligent systems for big data processing and decision support. This allows the opportunity to offer more informed solutions. The development of methods and models for multimodal data analysis leads to the creation of large language models (LLM) (Wang et al., 2024) and large multimodal models (LMM) (Li et al., 2024), which are used successfully when working with large sets of multimodal data.

In the modern world where innovation is so important, there is a significant increase in patent information associated with the intensive development of technologies and global digitalization. Thousands of new patents are published annually containing valuable data on technical innovations and industry trends. The patents themselves are an important source of information for companies, research organizations, and government agencies, because they help to track technological developments and to predict future directions for scientific research, industrial development and commercial

opportunity. However, a huge amount of this patent data is often analyzed and processed manually. This is a complex task that requires significant time and resource, and hence cost.

To solve these problems, and to create an effective tool, it is necessary to create automated systems for patent information analysis and processing. Automated systems that can process and analyze large amounts of patent data efficiently have the potential to reduce data processing time significantly and, in particular, improve the accuracy of forecasts regarding the emergence of new technological areas. The importance of developing such software systems is explained by the need for strategic planning and decision-making tasks that are based on reliable and timely information about technological trends (Berezkin et al., 2024). There are many approaches to patent data analysis and processing, which include clustering, networks research and machine learning, some of these approaches are described in Section II of this article.

In recent years the Natural Language Processing (NLP) and Deep Learning (DL) technologies have made significant progress, opening new opportunities for the creation of advanced and efficient patent analysis systems. One of the main tasks in this context is to predict the emergence of new technological areas, which requires the use of complex clustering and data classification algorithms.

This article describes a method for Predicting the Emergence of New Technological Areas based on multimodal patent information (PENTA). The PENTA method allows loading and processing data which is extracted from patent documents. PENTA converts the text information into a vector representation of the clusters of patents and visualizes the patents' landscape. The main emphasis of PENTA is on optimizing the clustering threshold and associating the obtained clusters with International Patent Classification (IPC) classes (Hoshino et al., 2023), thus improving the accuracy of forecasts and forming well-founded hypotheses about the emergence of new technologies.

## 2 RELATED WORKS

This section examines three main approaches to the patent information analysis from the point of view of the ability to predict the emergence of new / promising technological areas.

### 2.1 CLUSTERING-BASED APPROACH TO PATENT INFORMATION ANALYSIS

The clustering-based approach to patent information analysis aims to identify and predict new technological areas by grouping patents based on similar characteristics.

The first step of such analysis is to transform the textual patent information into numerical vectors. Traditional vectorization methods such as Term Frequency-Inverse Document Frequency (TF-IDF) (Qaiser & Ali, 2018) consider the frequency of word occurrence in documents only. The word embeddings models such as Word2Vec (Mikolov et al., 2013), GloVe (Ji et al., 2021), and FastText (Joulin et al., 2016) transform words into vectors (embeddings) reflecting their semantic meaning and context. The modern transformer models such as BERT (Devlin et al., 2019) or GPT-3 (Brown et al., 2020) can create deep contextual representations of text, which can account for more complex relationships between words and phrases. These models can be extended to the sentence and document level using Sentence-BERT (Reimers & Gurevych, 2019) or Universal Sentence Encoder (Sarkar et al., 2022).

The next step of such analysis involves the application of clustering algorithms for grouping patents: the algorithm K-means (MacQueen, 1967) divides data into K clusters minimizing the intra-cluster distance; hierarchical clustering (Nielsen, 2016) constructs a dendrogram, where clusters are formed by sequentially combining or dividing groups of data; the algorithm Density-Based Spatial Clustering of Applications with Noise (DBSCAN) (Ester et al., 1996) selects clusters based on the density of points in the vector space and allows an automatic determination of the number of clusters as well as separates off the noise data; the algorithm Affinity Propagation (Manoj et al., 2015) determines cluster centers by passing messages between data points, which allows for efficient grouping of large volumes of data.

An important aspect of the approach is the analysis of changes in the structure of clusters over time. The growth or shrinkage of clusters corresponds to increasing or decreasing interest in certain technological areas. The merger or separation of clusters corresponds to the combination of techno-

logical areas or emergence of new, specialized areas. The emergence of new clusters indicates the birth of new technological areas.

To help researchers and analysts to understand better and to interpret the complex relationships between patents and technology areas, various visualization tools are used: dendrograms which allow the researcher to see the structure and the connections between the clusters; heat maps which show a proximity between patents and clusters; network graphs which display the connections and interactions between the patents and the clusters.

It is important to mention that clustering-based methods can be used in combination in order to achieve the best results in predicting the new technology trends and identifying gaps in patent data.

## 2.2 RESOURCE-BASED APPROACH TO PATENT INFORMATION ANALYSIS

A resource network is a directed graph in which nodes can store resources (for example, patents), and edges have a certain capacity that limits the amount of resource transferred between nodes. At each discrete time step, the resource is redistributed between nodes observing the conservation law.

The main methods of the resource-based approach are: hidden Markov models (HMM) in combination with Bayesian networks (Jurafsky & Martin, 2023; Lee et al., 2017b); Markov processes for systems analysis (Aristodemou & Tietze, 2018); random walks and Markov chains (Vassiliou & Georgiou, 2021).

Each of these methods offers distinct advantages and limitations. Hidden Markov models and Bayesian networks excel at modeling complex dependencies and are effective for forecasting, but their primary drawbacks are the need for significant amounts of data and the complexity of their setup and interpretation. Markov processes are valued for their simplicity and power in modeling systems with discrete states; however, they are limited when dealing with more complex history dependencies and require fine-tuning of transition probabilities. Finally, random walks and Markov chains are highly adaptable for modeling network structures and accounting for temporal dynamics, though they can be computationally expensive for large networks and require careful parameter calibration. These methods facilitate a deep understanding of the dynamics of technology development and can serve as a basis for strategic planning in innovation activities and research.

## 2.3 MACHINE LEARNING APPROACH TO PATENT INFORMATION ANALYSIS

The general workflow for this approach has the following steps:

1. collecting the patent data, including description text, filing date, citations, and IPC classes;

2. cleaning data, processing text, vectorizing text information;

3. using historical data to train a classification model that predicts the emergence of new patent classes;

4. evaluating the accuracy of the model on test data, adjusting hyperparameters to improve the accuracy.

The main machine learning methods used for forecasting in the field of technological innovation are the following.

Gradient Boosting Decision Trees (GBDT) (Friedman, 2002). This method can be applied to patent data in order to identify trends and dependencies that indicate the possible emergence of new technological trends. For example, analyzing patent texts, their classifications, and timestamps can help in predicting future technological innovations (Li et al., 2020).

Random Forests (RF) (Breiman, 2001). This method can be applied to patent data in order to identify important features that influence the emergence of new technological areas. This information can then be used for accurate prediction (Santiago et al., 2021).

Multilayer Perceptron (MLP) (Alzubaidi et al., 2021; Popescu et al., 2009). The study (Lee et al., 2017a) used MLP to analyze patent data, which made it possible to identify new technologies at an early stage. The technique includes the use of various patent indicators and association rules for accurate and timely detection of developing technologies.

Convolutional Neural Networks (CNN) (Purwono et al., 2022). The article (Ji et al., 2024) discusses a technique for predicting the emergence of new technologies using CNN and MLP. The study uses 18 input and 3 output indicators from the US Patent Office database, making it possible to identify complex nonlinear relationships and analyze technology development trends at an early stage, which helps in strategic planning and forecasting technological trends.

Recurrent Neural Networks (RNN) (Das et al., 2023). The article (Ji et al., 2024) uses RNN and MLP to analyze patent data, which helps to accurately and promptly identify new technologies at early stages of their development.

Long Short-Term Memory (LSTM) (Staudemeyer & Morris, 2019). The article (Hou et al., 2024) uses LSTM to analyze the life cycle of technologies. The study proposes a new method for predicting the emergence of new technologies by analyzing patent data. The use of LSTM allows for more accurate and effective identification of technology development trends at different stages of their life cycle, from emergence to growth, maturity, and saturation.

Deep neural networks with Transformer architecture (Transformer) (Vaswani et al., 2023). The article (Zhao et al., 2024) focuses on identifying trends and selecting collaboration partners, taking in to account the interdependence of knowledge and collaboration. The study uses transformer models to analyze collaboration and knowledge networks, which allows organizations to strategically select partners for innovation projects and predict future technology trends. The use of transformers helps to analyze large amounts of data more accurately and improve the decision-making process, in turn helping to increase the efficiency of innovation.

Association Rule Mining (ARM) (Jun et al., 2012). The study (Sunghae, 2013) uses ARM and Box-Jenkins modeling methods to identify patterns in patent data, which allows the prediction of technology trends and the identification of new promising technologies. The study focuses on analyzing data from the European Patent Office using cluster analysis, text mining and social network methods, which allows for the formation of informative technology maps and improves strategic innovation planning.

Bayesian Networks (Peal, 1985). The study (Jeong et al., 2021) describes a methodology for creating technology roadmaps that not only adapt to changing circumstances, but also predict the emergence of new technologies. The authors of (Jeong et al., 2021) use Bayesian networks and topic modeling analysis to identify and assess risks, which allows for dynamically adjusting plans and predicting new technology trends. This machine learning approach has been successfully applied in the field of artificial intelligence, demonstrating its effectiveness and sustainability in technology management.

Each of the discussed methods has its own advantages and disadvantages, which makes them suitable for solving different types of problems: GBDT and Random Forest provide high accuracy and robustness to noise, but require significant computational resources and careful tuning; MLP and CNN are effective for analyzing data with nonlinear dependencies and features, but also require large amounts of data and computational costs; RNN and LSTM are especially useful for sequential data and time series, although they have problems with vanishing gradients; deep neural networks with Transformer architecture are a modern approach that provides high efficiency in natural language processing tasks, but also requires significant resources.

## 3 DATA USED FOR PATENT ANALYSIS

Patent documents are used as the input data for the development of the PENTA method. The documents were collected from the patent database of the United States Patent and Trademark Office (United States Patent and Trademark Office). When preparing the input data, the main goal was to collect a representative data set that would cover various technological areas and time periods. The data set includes patents published between 1995 and 2022. The patent information used for the research includes patent title, text description, filing date and IPC classes.

# 4 THE PROPOSED METHODS FOR PATENT LANDSCAPE MODELING

## 4.1 METHOD FOR CONSTRUCTING A PATENT LANDSCAPE MODEL

The method for constructing a patent landscape model includes several stages. These ensure the transformation of text data into vector representations, clustering, and parameter optimization. This method allows the creation of highly efficient models that can identify hidden patterns in patent data and predict the emergence of new technological areas.

First, patent texts (Title and Abstract) are transformed into vector representations using a language model. These vector representations are stored in the database and can be reused at a later time without the need for recalculation. There are language models that have been specifically adapted for analyzing patent documents. Some of these models are universal, trained on patent data from different domains. Examples: PatentBERT (Lee & Hsiang, 2019) and Patent Scientific BERT augmented (PatentSBERTa) (Bekamiri et al., 2024).

The PatentSBERTa and PatentBERT models demonstrate similar quality indicators (such as Precision, Recall and F1) when solving classification problems (Bekamiri et al., 2024). The preference for one model over another is determined by the characteristics of a particular problem being solved. The authors chose the PatentBERT model as it is best suited for analyzing the title, text description and IPC class of patents. This model is an adapted version of the BERT model for processing patent texts.

After the vector representations have been made for each patent, the clustering process is launched using the DBSCAN algorithm available at the scikit-learn library. The DBSCAN is a density algorithm well suited for identifying clusters of arbitrary shape and it is robust to noise in the data. It identifies groups of patents that have similar thematic and technological characteristics.

The quality of clustering is assessed using the Adjusted Rand Index (ARI) metric. The ARI measures the degree of correspondence between the clusters and the well-known IPC class labels. The ARI considers the random coincidence and provides a more accurate estimation in comparison to other metrics. During the assessment of the clustering quality, the clusters obtained at the first stage are analyzed and the correspondence between them and the IPC classes can be performed.

The DBSCAN algorithm can be adjusted to improve the quality of clustering and maximize the ARI metric. For this the gradient descent method allows the researcher to find the optimal value of the parameter "neighborhood size of a point" (that is, the maximum distance between two samples for one to be considered as in the neighborhood of the other) that maximizes the ARI metric. Optimization is carried out iteratively, which allows achieving the best division of patents into clusters.

This process includes a pairwise comparison of clustering accuracy and IPC labels, which allows determining how well the clusters match the classification. With high correspondence, cluster labels can be associated with IPC class labels, which simplifies further analysis and interpretation of the data.

After all stages are completed, the model is saved in the database. The clustering threshold, ARI metric, cluster labels and associated IPC class labels are included. This allows the model to be reused for analyzing new data and assessing its quality.

## 4.2 METHOD FOR CONSTRUCTING A PREDICTIVE MODEL

The method for building a forecast model is based on similar principles to the patent landscape model building method, but it includes additional steps accounting the time aspects in order to generate forecasts. This method allows the researcher to identify possible directions of technology development and potential new technology areas based on the analysis of changes in patent data over specific time periods.

In order to construct a predictive model, one needs to create a new dataset similar to the one used to build the base model, but covering a more recent time range. This new dataset is filtered using the same criteria as the original one to ensure comparability of the results. It is important that the IPC codes and other filtering parameters remain identical for both datasets.

Similar to the base model building method, the patent text data from the new dataset is transformed into vector representations using the PatentBERT model. The resulting vectors are stored in the database for subsequent use in the clustering and analysis process.

The DBSCAN method is used to cluster the data in the new dataset with the optimized parameter "neighborhood size of a point" found during the base model building step. This ensures consistency and allows comparison of clustering results across time periods. The DBSCAN method identifies groups of patents that have similar thematic and technological characteristics, which helps identify changes in the patent landscape.

After clustering, the quality of clustering is assessed using the Adjusted Rand Index (ARI) metric, similar to the baseline model. Next, the cluster labels are associated with the IPC class labels present in the filter of the new dataset. This helps determine how well the clusters reflect the classification of patents by IPC classes and identifies changes in thematic areas.

Based on the clustering results and the label association, a predictive patent landscape is created. To visualize changes in patent data, the Uniform Manifold Approximation and Projection (UMAP) is used, which allows building a two- or three-dimensional representation of clusters and their relationships. This landscape clearly shows how the technological areas have changed and where new clusters may appear.

After completing all stages, the predictive model is saved in the database. The clustering threshold, ARI metric, cluster labels and associated IPC class labels are included. This allows the model to be reused to analyze new data and assess its quality.

### 4.3 METHOD FOR ASSESSING THE DEVELOPMENT DIRECTIONS OF TECHNOLOGICAL AREAS

The method for assessing development directions is based on the analysis of changes in patent data and the identification of significant patterns using an association matrix. This process allows for the formation of hypotheses about possible directions of technological development and the assessment of the probability of their occurrence.

At the initial stage an association matrix is created, in which the X axis represents the clusters of the base model, and the Y axis represents the clusters of the forecast model. Each cell of the matrix contains the percentage of correspondence between the clusters. This indicator is calculated as the ratio of the number of patents present in both clusters to the total number of patents in the base cluster. This allows for the assessment of changes in the patent landscape and the identification of significant patterns.

The next step is the initialization of the list of hypotheses, which will be considered, based on the analysis of the association matrix. This list will contain possible development scenarios.

For each row in the association matrix, a model class (X-cluster) is determined. Then, for this row, cells with a match above a specified threshold are found. The number of such cells determines the further course of the analysis:

1. if there is more than one cell, a check is made to see if they belong to different classes (Y-clusters); if so, a hypothesis is created about the merging of classes and the probability of the emergence of a new technological area; otherwise, a hypothesis is created about diversification and a change in focus within the class;

2. if there is one cell, a hypothesis is created about maintaining the focus of the technological area;

3. if there are no cells and the overall compliance is less than the threshold, a hypothesis is created about a decrease in activity in this technological area.

After the hypotheses are formed, their probability is assessed. This assessment is based on an analysis of the degree of change in the association matrix, the number of patents and other indicators. Each type of hypothesis uses its own formula for calculating the probability: the hypothesis of class merging – the probability Ps is calculated based on the sum of the percentage correspondence of all

cells participating in the merger and the average growth rate of patents in these clusters:

$$P_s = \left( \frac{\sum_{i=1}^{n} C_i}{n} \right) \times G \tag{1}$$

where Ci is the percentage match for each cell, n is the number of cells, G is the average growth rate of patents; the hypothesis of diversification and change of focus – the probability Pd is based on the percentage of patents that have moved to a new cluster and the change in the topic of keywords identified using LDA:

$$P_d = \frac{P_m}{P_t} \times K \tag{2}$$

where Pm is the number of patents that have moved to the new cluster, Pt is the total number of patents in the original cluster, K is the coefficient of change of keywords; the hypothesis of a decrease in activity – the probability Pa is estimated based on the general decrease in activity in the cluster, calculated as the difference in the number of patents between the base and forecast periods:

$$P_a = 1 - \frac{P_p}{P_b} \tag{3}$$

where Pp is the number of patents in the forecast period, Pb is the number of patents in the base period.

At the final stage, the hypotheses and their probabilities are stored in the database. This allows decision makers to access the analyzed data and make informed decisions based on the forecasts received.

The method for assessing the development directions of technological innovations is the main element of the proposed approach, ensuring the identification of significant patterns and the formation of hypotheses about possible directions of technology development. The use of this method allows for the effective analysis of large volumes of data and provides users with valuable insights for making strategic decisions in the field of innovation and technology.

## 5 EXPERIMENTS

### 5.1 EXPERIMENTAL VERIFICATION OF THE PROPOSED METHODS

The patent data dated 2000 and 2010 has been selected for testing. The aim of the experiment was to compare the patent landscapes of 2000 and 2010 and to forecast technology trends based on this comparison.

To simplify the testing procedure, it was decided to narrow the analysis to several subclasses. The selected subclasses are large enough to provide sufficient data for analysis. They are also diverse enough to demonstrate the efficiency of the method in different areas. The authors decided to select the following subclasses: G05B, H01L, H04L and A61B. The selection process was performed before transferring the data to the classifier. After training the classifier on patent data for 2000, its accuracy was tested using cross-validation as shown in the figure. (Fig. 1). The cross-validation results show that the classifier was well trained.

Using the methods described in Section 4, patent landscape models were constructed for the selected subclasses (a base model based on 2000 data and a predictive model based on 2010 data). The visual representation of the models created using UMAP, is shown in Fig. 2 and Fig. 3 respectively.

Comparison of Fig. 2 and Fig. 3 suggests shifts in emphasis in patent documents (movement of cluster centroids and changes in keywords). Patent subclasses A61B and H04L have less in common, and patent class G05B has shifted to the side, which may also indicate the development of this cluster into an independent area. The proposed visualisation facilitated monitoring of centroid movements and changes in cluster boundaries, thereby allowing a comprehensive assessment of the dynamics of the patent landscape.

### 5.2 EVALUATING THE PERFORMANCE OF THE PROPOSED METHODS

In order to test the proposed approach for predicting the emergence of new technologies based on patent information, a series of 15 experiments were conducted on different datasets. Expert

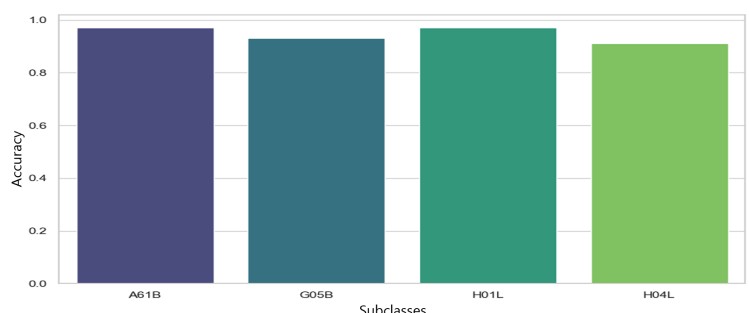

Figure 1: Classifier cross-validation results

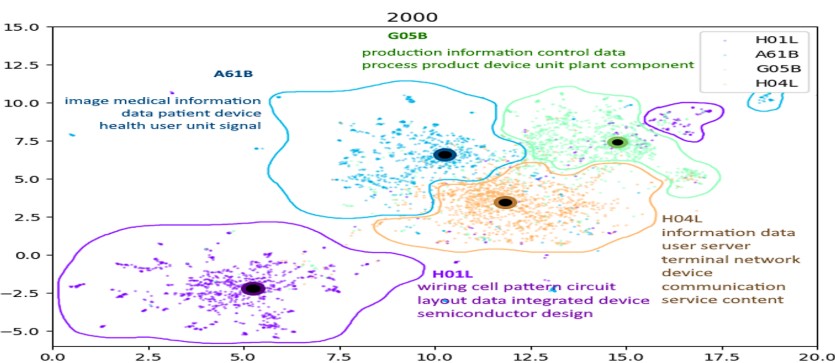

Figure 2: Distribution of patents in vector space for 2000

assessments were used to verify the accuracy and relevance of the predictions made via the proposed approach. In particular, one of the experiments analyzed the behavior of subclasses H01L, A61B, G05B, H04L in the period from 2010 to 2020. Comparison of data between 2010 and 2020 showed that the content of patents in G05B (regulatory and control systems) has become significantly closer to patents in the areas of medical solutions in diagnostics and surgery (A61B) and infrastructure systems for transmitting digital information (H04L). This indicates an increase in the number of solutions at the intersection of these areas. This trend is expected to intensify in the coming years due to the synergetic effect of the introduction of artificial intelligence and neuromorphic computing in the medical domain. Thus, it is likely that patent classes G05B and H04L will continue to converge, and at their intersection a new class dedicated to intelligent medical solutions may emerge.

To evaluate the accuracy of the proposed methods, an accuracy indicator was used, calculated using the following formula:

$$\text{Accuracy} = \frac{\text{Number of correct predictions}}{\text{Total number of predictions}} \tag{4}$$

In this case, correct predictions are those where the outcome produced via the proposed approach matches the expectations of experts.

Of the 15 scenarios, 11 were marked as correct predictions and 4 as incorrect predictions:

$$\text{Accuracy} = \frac{11}{15} \approx 73.3\% \tag{5}$$

The testing results showed that the forecasting method is able to correctly identify the main directions of technology development in most cases. Expert assessment confirmed the high accuracy and relevance of the predictions.

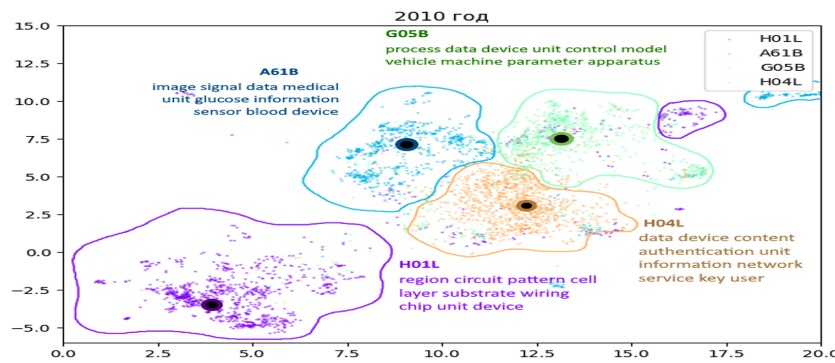

Figure 3: Distribution of patents in vector space for 2010

## 6 CONCLUSION AND FUTURE WORKS

The article considers the challenge of developing a new approach to analyzing and forecasting new technological areas based on patent information (PENTA). The proposed approach allows for efficient analysis of large arrays of patent data and identification of potential directions of technological development, which is critical for strategic planning and innovation.

The article considered various approaches and algorithms used for patent data analysis, including machine learning and natural language processing (NLP) methods. The capabilities and limitations of each method were studied, which made it possible to select the optimal solutions to develop a novel approach.

The authors developed methods for constructing a patent landscape model and a forecast model, as well as a method for assessing development directions in technological areas. Particular attention was paid to algorithms for optimizing the clustering threshold and forming associations between patents and technological directions.

The proposed methods uses the PatentBERT model to transform patent texts into a vector representation, the DBSCAN method for patent clustering, and the Adjusted Rand Index metric to optimize the clustering threshold. UMAP and LDA methods are used to visualize the patent landscape, providing a clear observation of the structure and relationships of the data.

The next steps will be to develop an understanding of model and forecasting accuracy more deeply by applying the tools to a wider range of technology areas. In a world where the cost of technological innovation is high, opportunities to better understand where to focus, capitalize on advancements and 'break new ground' can be of value to research, government and business organizations in many ways. An informed approach to research and innovation is a fundamental element of successful foresighting.

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
