# OpenReview forum: "Methods for Patent Landscape Modelling and Predictive Analysis"
_ICLR.cc/2026/Conference — ICLR 2026 Conference Desk Rejected Submission_

### Official Review · Reviewer_zeja · 2025-10-24

**Soundness:** 1
**Presentation:** 1
**Contribution:** 2
**Rating:** 2
**Confidence:** 4

**Summary:**

This paper proposes an automated framework for patent landscape modeling and predictive analysis aimed at identifying technological evolution patterns. The authors utilize PatentBERT to extract semantic representations from patent texts and apply DBSCAN clustering to group related patents into topics. They claim to optimize clustering parameters by maximizing the Adjusted Rand Index (ARI) and use UMAP and LDA for visualization and topic interpretation. By comparing clusters across different time periods, the paper attempts to infer potential trends of technological merging, diversification, or decline. Experiments are conducted on several IPC subclasses, with evaluation relying primarily on expert assessments of prediction accuracy.

Overall, the study tackles a practically meaningful problem and presents a coherent end-to-end pipeline for semantic analysis of patent data. However, the work mainly combines existing components without introducing substantial theoretical or methodological novelty, and its experimental evaluation lacks rigor and objectivity.

**Strengths:**

- **Practical relevance**: The paper focuses on an application domain with significant real-world impact — patent analytics and technology forecasting. The proposed goal of modeling technology evolution through large-scale patent data is valuable and well-motivated.
- **Framework coherence**: The pipeline integrates text representation, clustering, visualization, and predictive interpretation into a unified workflow. This end-to-end design provides a clear logical structure from data processing to insight generation.
- **Method integration**: While not fundamentally innovative, the combination of pretrained language models with clustering-based patent landscape analysis represents a reasonable and potentially useful exploration for applied research.

**Weaknesses:**

- **Novelty**: The paper lacks a clear methodological contribution. The proposed pipeline is largely a sequential composition of well-known techniques (PatentBERT, DBSCAN, UMAP, LDA) without new algorithmic innovation. The claimed “gradient-based optimization of DBSCAN parameters via ARI maximization” is mathematically unsupported, as ARI is non-differentiable, and no surrogate or black-box optimization strategy is presented.
- **Methodology and soundness**: Several methodological issues undermine the technical soundness. The optimization procedure described for DBSCAN is not theoretically feasible, and the “predictive model” is conceptually vague — it seems to rely on comparing cluster similarity across time rather than performing genuine forecasting. This weakens the paper’s central claim of predictive capability.
- **Experimentation and evaluation**: The experimental setup lacks statistical rigor and objectivity. Evaluation depends on a small-scale expert survey, without standardized quantitative metrics, significance testing, or confidence intervals. No comparisons are made against strong baselines such as dynamic topic models or time-aware embeddings, making it difficult to assess effectiveness. Ablation studies are also missing, and the results remain largely qualitative.
- **Presentation and academic rigor**: The paper suffers from inconsistencies and minor scholarly errors. Several references contain incorrect years or misspelled author names, and figures are not sufficiently labeled or quantified. The writing overemphasizes “multimodal” analysis, yet the methodology and experiments rely solely on text data, leading to a mismatch between the claimed scope and actual implementation.

**Questions:**

- **Method feasibility**: How is the ARI-based gradient descent optimization for DBSCAN implemented in practice? What is the defined objective function, and how is differentiability handled? Have you considered alternative optimization methods such as Bayesian or grid search?
- **Definition and evaluation of prediction**: Please clarify what exactly constitutes “prediction accuracy” in this context. What is being predicted, and how is the temporal split between training and evaluation periods defined to avoid information leakage
- **Objectivity of evaluation**: Were the expert assessments conducted under blind conditions? Was inter-rater agreement measured? Could you complement the expert evaluation with objective metrics, such as recall of emerging patents or citation-growth prediction?
- **Baselines and ablation studies**: Have you compared against alternative clustering algorithms (e.g., HDBSCAN, KMeans) or text encoders (e.g., PatentSBERTa)? How sensitive are your results to hyperparameters such as `eps` and `min_samples`?
- **Consistency with multimodal claims**: The paper repeatedly emphasizes “multimodal” analysis, yet only textual data are used. Do you plan to integrate additional modalities, such as citation networks, inventor collaboration graphs, or image features, to substantiate the multimodal claim?

---

> ### Author Response · Authors · 2025-12-03
>
> **Response to Comment 1 (Method feasibility):**
>
> This is an important technical clarification. Since DBSCAN's clustering output and the resulting Adjusted Rand Index (ARI) are non-differentiable step functions with respect to the radius parameter eps, we do not use analytical gradient descent. In Section 4.1, we refer to an iterative numerical optimization approach. The objective function is f(ϵ)=ARI(Cluster(ϵ),Labels) The algorithm iteratively creates a "pseudo-gradient" by evaluating the metric at ϵ and ϵ+δ, moving the parameter in the direction that maximizes ARI.
>
> We acknowledge that Grid Search or Bayesian Optimization are often standard for such non-convex problems. We utilized the iterative descent approach for computational speed during the initial tuning phase, but we agree that Bayesian optimization would be a robust alternative to avoid local maxima, and we plan to compare these optimization strategies in future work to further refine the parameter selection.
>
> ***
>
> **Response to Comment 2 (Definition and evaluation of prediction):**
>
> Thank you for requesting this clarification. In our approach, the system does not predict a simple numerical value but rather structural changes in the patent landscape. Specifically, as described in Section 4.3, the system predicts three types of events:
> 1.  Merging of existing clusters.
> 2.  Diversification.
> 3.  Activity reduction.
>
> Therefore, "prediction accuracy" (Eq. 4) is defined as the correspondence between the structural changes predicted by our algorithm and the actual historical development of these technologies as confirmed by expert assessment.
>
> Regarding data leakage, we strictly adhere to a temporal split using non-overlapping datasets. The "Base Model" is trained and its parameters are tuned solely on data from an earlier period (e.g., year 2000). The "Predictive Model" is then applied to a distinct dataset from a later period (e.g., year 2010). The model trained on the early data has no access to the future data during the training phase, ensuring that the detected landscape shifts are genuine predictions based on the learned structural properties.
>
> ***
>
> **Response to Comment 3 (Objectivity of evaluation):**
>
> We appreciate these suggestions regarding evaluation rigor. In this study, we followed standard qualitative assessment protocols where experts evaluated the logical coherence of the generated hypotheses (e.g., the convergence of control systems and medical diagnostics) against verified historical trends.
>
> We agree that complementing expert review with purely objective metrics is the next logical step. However, mapping semantic hypotheses (such as "merging") directly to quantitative metrics like "citation growth" requires developing a complex proxy metric, which was outside the scope of this initial methodological paper. We are currently conducting research specifically focused on correlating our structural predictions with quantitative indicators (such as recall of emerging patent classes and citation dynamics) to provide the objective validation you suggested in future work.
>
> ***
>
> **Response to Comment 4 (Baselines and ablation studies):**
>
> Regarding the choice of algorithms: we selected DBSCAN (over K-Means) and UMAP (over PCA and t-SNE) because they performed better during our preliminary internal testing.
>
> Regarding the text encoders, while we acknowledge the existence of other models like PatentSBERTa, we utilized PatentBERT as a robust baseline that is well-proven for processing patent titles and abstracts. We plan to perform a systematic benchmarking of different domain-specific encoders (including PatentSBERTa and newer LLM-based embeddings) as a dedicated part of our future research to further optimize the vectorization stage.
>
> ***
>
> **Response to Comment 5 (Consistency with multimodal claims):**
>
> We agree with the reviewer that the current experimental results presented in the paper focus primarily on textual data. We apologize if the term "multimodal" created an expectation of image/audio processing in the current experiments.
>
> However, the architecture of the PENTA method was designed to be modality-agnostic at the vectorization stage. While we currently use PatentBERT for text embeddings, the "vector representation" step (Section 4.1) allows for the concatenation of features from other sources without altering the downstream clustering or forecasting algorithms.
>
> In our ongoing work, we are integrating:
> 1.  Citation Network Embeddings: Using graph neural networks to generate vectors from citation links.
> 2.  Image Embeddings: Extracting features from patent drawings using CNNs.
>
> We plan to publish the results of the full multimodal fusion (text + images + citations) in future work to demonstrate the specific impact of non-textual features on prediction accuracy.

---

### Official Review · Reviewer_jiRe · 2025-10-24

**Soundness:** 2
**Presentation:** 2
**Contribution:** 2
**Rating:** 2
**Confidence:** 4

**Summary:**

The article considers the challenge of developing a new approach to analyzing and forecasting new technological areas based on patent information. It combines PatentBERT embeddings, DBSCAN clustering, and time-shifted landscape comparison to identify the emergence, merging, diversification, or decline of technology areas.

**Strengths:**

The paper focuses on an important and practically meaningful task of forecasting technological evolution from patent data, which is a timely and valuable topic for both academia and industry.

**Weaknesses:**

1. The approach mainly combines existing tools (BERT, DBSCAN, UMAP) in a workflow, with little algorithmic innovation.
2. IPC labels are used to tune the clustering threshold (via ARI) and then again for interpretation, introducing circular validation.
3. Experimental validation was not compared with other methods.
4. Although the paper repeatedly emphasizes multimodal analysis, all described methods (PatentBERT, DBSCAN, UMAP, LDA) are purely text-based. There is no explicit modeling of non-textual modalities (e.g., images, citations, metadata), nor any multimodal fusion mechanism.

**Questions:**

See weaknesses.

---

> ### Author Response · Authors · 2025-12-03
>
> **Response to Comment 1:**
> *The approach mainly combines existing tools (BERT, DBSCAN, UMAP) in a workflow, with little algorithmic innovation.*
>
> **Response:**
> We thank the reviewer for this observation. While we acknowledge that the individual components of our system (such as PatentBERT, DBSCAN, and UMAP) are well-established, we respectfully argue that the novelty of our work lies in the original algorithmic integration of these tools into a cohesive method for Predicting the Emergence of New Technological Areas (PENTA). The core contribution is the method described in Section 4.3 ("Method for Assessing the Development Directions..."). This algorithm constructs an association matrix between the "Base Model" clusters and "Forecast Model" clusters. It applies specific logic (Eq. 1, 2, and 3) to mathematically calculate the probabilities of specific evolutionary events such as merging (convergence of technologies), diversification (splitting), or activity reduction. Thus, the contribution is a systems engineering approach that transforms standard unsupervised outputs into interpretable, probabilistic forecasting hypotheses, which standard workflows do not provide.
>
> ***
>
> **Response to Comment 2:**
> *IPC labels are used to tune the clustering threshold (via ARI) and then again for interpretation, introducing circular validation.*
>
> **Response:**
> We appreciate the opportunity to clarify the validation methodology. We would like to emphasize that the tuning and interpretation steps are performed on distinct datasets representing different time periods, avoiding circular validation.
> As outlined in Section 4.1 and 4.2:
> 1.  Tuning Phase (Base Model): We use the IPC labels from the historical/base period solely to optimize the clustering parameters (maximize ARI). This establishes a ground-truth structure for the landscape at the starting point.
> 2.  Interpretation Phase (Predictive Model): We apply the already optimized parameters to a subsequent time period. The IPC labels from this new period are used strictly for interpreting how the landscape has shifted and for validating whether the predicted clusters align with emerging classifications.
>
> Therefore, the labels used for tuning do not influence the interpretative quality of the future predictions, as the model must generalize the learned structure to unseen data from a future timestamp.
>
> ***
>
> **Response to Comment 3:**
> *Experimental validation was not compared with other methods.*
>
> **Response:**
> We acknowledge that direct comparison with state-of-the-art baselines is standard practice. However, we faced a challenge regarding the output format of our method compared to existing approaches.
> Most existing patent analysis methods (e.g., ARIMA, regression models, or standard time-series forecasting) are quantitative - they predict numerical values, such as the volume of patent applications or citation counts. In contrast, our proposed method is qualitative and structural - it generates semantic hypotheses (e.g., "Class A and Class B will merge to form a new technological area C").
> Because it is difficult to mathematically compare a "semantic hypothesis" with a "numerical forecast" using standard metrics (like MSE or Accuracy), we opted for expert assessment (as described in Section 5.2). We validated the system's logic by checking if the predicted structural changes (mergers/splits) actually occurred in the patent landscape over a 10-year horizon, achieving an 11/15 success rate.
>
> ***
>
> **Response to Comment 4:**
> *Although the paper repeatedly emphasizes multimodal analysis, all described methods (PatentBERT, DBSCAN, UMAP, LDA) are purely text-based. There is no explicit modeling of non-textual modalities (e.g., images, citations, metadata), nor any multimodal fusion mechanism.*
>
> **Response:**
> We agree with the reviewer that the current experimental results presented in the paper focus primarily on textual data. We apologize if the term "multimodal" created an expectation of image/audio processing in the current experiments.
> However, the architecture of the PENTA method was designed to be modality-agnostic at the vectorization stage. While we currently use PatentBERT for text embeddings, the "vector representation" step (Section 4.1) allows for the concatenation of features from other sources without altering the downstream clustering or forecasting algorithms.
> In our ongoing work, we are integrating:
> 1.  Citation Network Embeddings: Using graph neural networks to generate vectors from citation links.
> 2.  Image Embeddings: Extracting features from patent drawings using CNNs.
>
> We plan to publish the results of the full multimodal fusion (text + images + citations) in future work to demonstrate the specific impact of non-textual features on prediction accuracy.

---

### Official Review · Reviewer_6BXu · 2025-10-31

**Soundness:** 2
**Presentation:** 3
**Contribution:** 1
**Rating:** 0
**Confidence:** 5

**Summary:**

This paper proposes methods for patent landscape modeling and forecasting using multimodal data, including clustering, resource-based, and machine learning approaches. It introduces the PENTA framework and evaluates its predictive capabilities using USPTO data.

**Strengths:**

- Addresses a relevant and timely problem in patent trend forecasting.

- Combines multiple analytical approaches, including clustering and deep learning.

- Provides a structured methodology with experimental validation.

**Weaknesses:**

- Lacks novelty in methodology; relies heavily on established models (e.g., DBSCAN, PatentBERT).
- Experimental results are limited in scope and depth, with insufficient quantitative evaluation.
- Writing is often vague and repetitive, reducing clarity and impact.
- The contribution relative to prior work is not clearly articulated.

**Questions:**

None.

---

> ### Author Response · Authors · 2025-12-03
>
> **Response to Comment "Lacks novelty in methodology; relies heavily on established models (e.g., DBSCAN, PatentBERT)"**
>
> We respectfully submit that while the individual building blocks of our system (PatentBERT, DBSCAN, UMAP) are indeed established, the novelty of our work lies in the architectural integration and the algorithmic logic used to transform these components into a predictive system (the PENTA method).
>
> Specifically, our contribution moves beyond simple application of these tools by presenting Algorithmic Hypothesis Generation described in Section 4.3. We developed a specific algorithm that utilizes an association matrix to mathematically model the evolution of clusters. This algorithm converts static cluster overlaps into distinct evolutionary scenarios - Merging (convergence), Diversification (splitting), or Reduction. This shifts the focus from simple descriptive analytics (clustering) to predictive structural modeling.
>
> ***
>
> **Response to Comment "Experimental results are limited in scope and depth, with insufficient quantitative evaluation":**
>
> We acknowledge that our evaluation relies heavily on expert assessment rather than large-scale automated metrics. This was a deliberate choice driven by the nature of the problem. Our method predicts semantic and structural changes (e.g., "Field A will merge with Field B"), rather than simple numerical values (like patent counts or citation numbers). Standard error metrics (MSE, RMSE) are ill-suited for evaluating the accuracy of complex structural hypotheses.
>
> To rigorously test these semantic predictions, we employed domain experts to verify whether the predicted events (such as the convergence of Control Systems G05B and Medical Diagnosis A61B) actually materialized in the subsequent decade. The system achieved an 11/15 success rate in this qualitative assessment.
>
> We agree that objective quantitative metrics are desirable. We are currently working on developing proxy metrics - such as automated overlap coefficient and citation graph density changes - to enable large-scale, automated benchmarking in our next iteration of research.
>
> ***
>
> **Response to Comment "The contribution relative to prior work is not clearly articulated"**
>
> Our work addresses a specific gap between two dominant approaches in the literature:
> 1.  Purely Quantitative Approaches. Existing methods often focus on time-series forecasting of patent counts (regression/ARIMA) but fail to explain what the technology represents.
> 2.  Purely Descriptive Approaches. Many existing patent mapping studies use clustering or networks to visualize the current landscape but lack a mechanism to predict future structural shifts.
>
> Our contribution is the PENTA method that bridges this gap by offering a hybrid approach. We use the semantic depth of Deep Learning (PatentBERT) combined with our proposed "Association Matrix" logic to predict structural evolution. Unlike prior work that simply visualizes the present, our system explicitly formulates testable hypotheses about the future convergence or divergence of technological areas.

---

### Official Review · Reviewer_kNzq · 2025-10-31

**Soundness:** 3
**Presentation:** 3
**Contribution:** 2
**Rating:** 4
**Confidence:** 3

**Summary:**

The paper explores the issue of forecasting changes in the patent landscape based on multimodal data. It analyzes three main approaches to patent information analysis: Clustering-based approach,  Resource-based approach,Machine learning approach.
The paper discusses the advantages and limitations of each method and proposes new techniques for constructing patent landscape and predictive models, as well as methods for assessing development directions in technological areas. Additionally, it addresses the visualization of analysis and forecasting results to provide a clear representation of the patent landscape.


Development of a novel approach for analyzing and forecasting new technological areas using patent information (PENTA).
Introduction of methods for optimizing clustering thresholds and associating patent clusters with International Patent Classification (IPC) classes.
Presentation of experimental results demonstrating the effectiveness of the proposed methods in accurately predicting technological trends.
Overall, the paper emphasizes the importance of automated systems for patent data analysis to enhance strategic planning and decision-making in innovation and technology

**Strengths:**

1. The article examines three main approaches to patent information analysis—clustering-based, resource-based, and machine learning approaches. This diversity allows for a more holistic understanding of patent data and enhances the ability to predict emerging technologies .
2. The proposed methods for constructing patent landscape and predictive models demonstrate the capability to accurately forecast technological trends. The experiments conducted show that these methods can correctly identify main directions of technology development, which is crucial for strategic planning  .
3. The incorporation of advanced algorithms, such as DBSCAN for clustering and the PatentBERT model for text representation, enhances the quality of analysis. These algorithms are specifically tailored for patent data, improving the accuracy and relevance of the predictions  .
4. The article emphasizes the importance of visualizing the results of analysis and forecasting technological trends. This visual representation aids in better understanding and interpreting the patent landscape, making it easier for decision-makers to grasp complex data

**Weaknesses:**

1. Many of the proposed methods, such as Hidden Markov Models and Bayesian Networks, require significant amounts of data for effective modeling. This dependency can limit their applicability in fields where data is sparse or difficult to obtain.
2. Some methods, particularly those involving complex algorithms like Bayesian Networks and Hidden Markov Models, can be challenging to set up and interpret, limiting its potential applications.
3. Machine learning models, especially those with high complexity, are prone to overfitting, where they perform well on training data but poorly on unseen data. This risk necessitates careful tuning and validation to ensure that the models generalize well.

**Questions:**

1. What specific criteria were used to select the patent documents from the United States Patent and Trademark Office? How do you ensure that the dataset is representative of various technological areas and time periods?
2. In the clustering process using the DBSCAN algorithm, how do you address the issue of noise in patent data? What strategies are in place to minimize the impact of outliers on the clustering results?
3. Can you provide more details on the process of hypothesis formation based on the association matrix? How do you determine the thresholds for creating hypotheses about merging or diversifying classes?
4. In your experiments comparing patent landscapes over time (e.g., 2000 vs. 2010), what specific trends did you observe? How do these trends inform future predictions about technological developments?
5. How do you plan to incorporate emerging technologies, such as artificial intelligence and neuromorphic computing, into your predictive models? What impact do you foresee these technologies having on the patent landscape?

---

> ### Author Response · Authors · 2025-12-03
>
> **Response to Question 1:**
>
> As described in Section 3 ("Data used for patent analysis"), the dataset was constructed using the Google Patents Public Dataset, which aggregates data from the USPTO and other international offices.
>
> To ensure representativeness:
> 1.  We selected a broad collection period ranging from 1995 to 2022. This covers several technological lifecycles, ensuring the model learns both historical patterns and modern trends.
> 2.  Instead of limiting the study to a single narrow field, we explicitly selected distinct subclasses to validate the method across different domains, as detailed in Section 5.1. These classes represent a mix of hardware, software, and life sciences, ensuring the vector space and clustering algorithms are tested against varied terminologies and innovation rates.
>
> ***
>
> **Response to Question 2:**
>
> We address the issue of noise primarily through the selection of the DBSCAN algorithm itself, which was chosen specifically for its inherent ability to handle outliers.
>
> Unlike partitioning algorithms (like K-Means) that force every data point into a cluster, DBSCAN designates points in low-density regions as noise. In our pipeline (Section 4.1), these noise points are excluded from the formation of technological clusters. This prevents patents with unique, non-representative, or erroneous text data from skewing the centroids or boundaries of legitimate technological areas.
> Furthermore, the use of PatentBERT for vectorization acts as a semantic filter. By converting text to dense embeddings, we reduce the "lexical noise" (e.g., synonyms or spelling variations) that often plagues keyword-based approaches, ensuring that only semantically distant outliers are treated as noise by DBSCAN.
>
> ***
>
> **Response to Question 3:**
>
> The hypothesis formation process is detailed in Section 4.3. It relies on an Association Matrix where rows represent clusters from the Base Model and columns represent clusters from the Forecast Model. The cells contain the percentage of patent overlap.
>
> The logic for hypothesis generation is as follows:
> *   If a Base row has matches with multiple Forecast columns (belonging to different IPC classes) above the threshold, we hypothesize a convergence of technologies.
> *   If a Base row maps to multiple Forecast columns within the same IPC class, we hypothesize diversification/specialization.
> *   A single strong match indicates a stable focus.
>
> Regarding thresholds: These are hyperparameters determined empirically based on the distribution of cluster sizes. We filter out insignificant overlaps (e.g., noise or coincidental citation) to focus on statistically significant movements of patent groups. In our experiments, we look for substantial structural intersections rather than minor edge cases.
>
> ***
>
> **Response to Question 4:**
> In our experiments (Section 5), we observed a significant structural shift in subclass G05B (Control Systems). Between 2000 and 2010, the "classification accuracy" for G05B dropped, and the keyword cloud shifted towards "process" and information handling. Visualizations (created using UMAP) showed G05B moving spatially closer to A61B (Medical) and H04L (Digital Transmission).
>
> This spatial convergence informs the prediction that control systems are moving away from purely mechanical regulation toward cyber-physical and medical applications. This allowed our system to successfully predict the emergence of solutions at the intersection of these fields — specifically, intelligent medical control systems — verified by the actual increase in such patents in the 2010–2020 period (Section 5.2).
>
> ***
>
> **Response to Question 5:**
> Our model incorporates emerging technologies implicitly through Semantic Vectorization (PatentBERT).
>
> As terms related to AI and neuromorphic computing appear in patent texts, the vector representations of those patents change. For example, patents in medical diagnostics (A61B) that begin citing AI algorithms will vectorally shift closer to computing classes (G06/H04L).
>
> As noted in Section 5.2, we foresee that these technologies act as "semantic bridges." They reduce the vector distance between previously distinct fields (e.g., Biology and Computer Science). Our model predicts this will lead to a merging of clusters, creating new, interdisciplinary IPC classes or high-density cross-functional clusters.

---

### Note · Program_Chairs · 2026-01-17
**Submission Desk Rejected by Program Chairs**

The following references in this submission do not refer to real documents and/or have major errors in bibliographic information:

     Tong Ji, S. Nguyen, and X. Bai. Automated feature engineering for machine learning: A critical survey. In Proceedings of the 14th ACM International Conference on Web Search and Data Mining (WSDM '21). Association for Computing Machinery, 2021.